# Neurologic Outcome Comparison between Fetal Open-, Endoscopic- and Neonatal-Intervention Techniques in Spina Bifida Aperta

**DOI:** 10.3390/diagnostics13020251

**Published:** 2023-01-09

**Authors:** Deborah A. Sival, Agnieszka Patuszka, Tomasz Koszutski, Axel Heep, Renate J. Verbeek

**Affiliations:** 1Department of Pediatric Neurology, Beatrix Children’s Hospital, University Medical Center Groningen, University of Groningen, 9700 RB Groningen, The Netherlands; 2Department of Gynaecology, Obstetrics and Oncological Gynaecology, School of Medicine with Division of Dentistry in Zabrze, Medical University of Silesia, 40-055 Katowice, Poland; 3Department of Pediatric Surgery and Urology, Faculty of Medical Sciences in Katowice, Medical University of Silesia, 40-055 Katowice, Poland; 4Department of Pediatrics and Research Center Neurosensory Science, Carl von Ossietzky University Oldenburg, 26129 Oldenburg, Germany; 5Department of Pediatric Neurology, Emma Children’s Hospital, Amsterdam University Medical Centre, 1105 AZ Amsterdam, The Netherlands

**Keywords:** Spina bifida, fetal intervention, muscle ultrasound, child, neurologic assessment

## Abstract

Introduction: In spina bifida aperta (SBA), fetal closure of the myelomeningocele (MMC) can have a neuroprotective effect and improve outcomes. In Europe, surgical MMC closure is offered by fetal-open (OSBAR), fetal-endoscopic (FSBAR), and neonatal (NSBAR) surgical techniques. Pediatric neurologists facing the challenging task of counseling the parents may therefore seek objective outcome comparisons. Until now, such data are hardly available. In SBA, we aimed to compare neurologic outcomes between OSBAR, FSBAR, and NSBAR intervention techniques. Methods: We determined intervention-related complications, neuromuscular integrity, and neurologic outcome parameters after OSBAR *(n* = 17) and FSBAR (*n* = 13) interventions by age- and lesion-matched comparisons with NSBAR-controls. Neurological outcome parameters concerned: shunt dependency, segmental alterations in muscle ultrasound density (reflecting neuromuscular integrity), segmental motor-, sensory- and reflex conditions, and the likelihood of intervention-related gain in ambulation. Results: Compared with NSBAR-controls, fetal intervention is associated with improved neuromuscular tissue integrity, segmental neurological outcomes, reduced shunt dependency, and a higher chance of acquiring ambulation in ≈20% of the operated children. Children with MMC-lesions with a cranial border at L3 revealed the most likely intervention-related motor function gain. The outcome comparison between OSBAR versus FSBAR interventions revealed no significant differences. Conclusion: In SBA, OSBAR- and FSBAR-techniques achieved similar neuroprotective results. A randomized controlled trial is helpful in revealing and compare ongoing effects by surgical learning curves.

## 1. Introduction

In fetal spina bifida aperta (SBA), it is well-known that exposure of the myelomeningocele (MMC) to the intra-uterine environment can lead to ongoing neurological damage (the 2nd hit of damage) [1,2], which can be ameliorated by fetal neuroprotective strategies [3]. Almost a decade ago, the Management of Myelomeningocele Study (MOMS) provided the first convincing evidence in human SBA fetuses that open, extra-uterine spina bifida repair (OSBAR) could reduce the number of shunt placements and improve motor outcome [4]. Since then, many centers have adopted the OSBAR approach [4] by using the same standardized skills and techniques [5,6]. But the OSBAR approach is considered an invasive technique, potentially leading to iatrogenic morbidity [7]. This has led to the development of the intra-uterine “fetoscopy key-hole” technique (FSBAR), striving for a less invasive technique [8]. However, as we have previously shown, the initial application of this technique appeared at the high cost of intervention-related morbidity, preterm birth, and even mortality [5,9,10]. It was decided to await further surgical improvement before advocating widespread implementation [10,11].

In Europe, both FSBAR and OSBAR techniques are being performed in clinical settings. Neurological outcomes of these techniques are usually expressed as a “segmental neurologic gain.” This outcome measure comes from comparing the expected neurological function (based on the morphological upper level of the MMC) with the observed neurologic function (based on the postnatal pediatric neurologic examination). However, as no fetus with spina bifida (SBA) is the same as the other (for instance, regarding the variety in cerebral pathology, morphology of the spinal cord and neural tube defect [2], and clinical complications), multiple factors can contribute to the actual neurological examination after birth. This is not only attributable to inter-individual morphological heterogeneity but also to the heterogeneity of intervention-related morbidity [8,10], including the effects from surgical learning curves [12,13]. Thus, even when the average effect of fetal intervention is based on a large group of patients, one cannot generalize the average group results to the individual SBA patient. Pediatric neurologists facing the challenging task of counseling parents (expecting a child with SBA) may therefore seek more objective quantitative information so that parents can anticipate the decision that suits their individual situation best. The targeted parameters for such a “best decision” may differ among parents, as some may strive for the least intervention-related health risks and hazards, some may strive to prevent shunt implantation and preserve cognition, whereas others may strive for the best neurologic motor outcome including the highest chance to acquire ambulation. For the latter goal, one would need information on the prevention or amelioration of the 2nd hit of damage per fetal intervention technique. Until now, however, objective quantitative data comparing this parameter between FSBAR, OSBAR, and NSBAR techniques [7,14] are scarce.

In this perspective, we applied the muscle ultrasound technique. Previously, we have shown that the influence of the MMC on muscle integrity (reflecting the 2nd hit of damage) can be quantified by calculating the intra-individual difference in muscle ultrasound density between spinal segments caudal- versus cranial- to the MMC, i.e., the dMUD value [10,15]).

By the application of this dMUD parameter, one can thus use each child as its own control and (partly) avoid the confounding influence of SBA-related heterogeneity (see Figure 1).

In the present study, we thus aimed to compare dMUD values between children operated by FSBAR and OSBAR techniques after matching each child with an NSBAR- control with the same MMC level at a comparable age. Furthermore, we aimed to explore the dMUD outcomes for the functional significance of motor function.

This study was conducted by the active collaboration between three experienced European centers, each reporting on a different SBA intervention technique for more than a decade (OSBAR [16,17]; FSBAR [10,18,19] and NSBAR [10,20,21]).

## 2. Methods

The medical ethics committees of Bonn University, Germany, Medical University of Silesia, Poland, and the University Medical Center Groningen (UMCG), the Netherlands, approved the present study. All parents of the children included gave informed consent.

### 2.1. Patient Data

#### 2.1.1. Patient Inclusion

Inclusion of fetally operated children: All surviving children that had previously received endoscopic fetal treatment between 2003 and 2009 at Bonn were invited to be included. Parents of all 13/13 FSBAR children consented to participate. The dMUD results in these 13 FSBAR children and their matched pairs with age- and lesion-matched NSBAR controls have been published [10]. Subsequently, all surviving children that received open fetal treatment between 2011 and 2017 at Katowice (*n* = 32) were invited, both by mail and by phone call. Parents of 17/32 OSBAR children consented to participate. Reasons for declined invitations were related to patient-bound factors (including no response, illness, journey issues, or other obligations).

Inclusion of neonatally operated children (NSBAR): All children from the NSBAR group were included as controls in age- and lesion-matched pairs with the children from the FSBAR or OSBAR groups. This resulted in 2 groups of 13 and 17 matched pairs (FSBAR and OSBAR, respectively). When more than one age- and lesion-matched NSBAR child was available for a match, we included the first match from the database. 

For outcome comparison, we thus obtained two groups of age- and MMC-matched pairs: 1. OSBAR versus NSBAR (for patient data, see Table 1) and 2. FSBAR versus NSBAR (for patient data, see [10]). Medians and ranges of age- and lesion- matched FSBAR and OSBAR children, see Table 2.

#### 2.1.2. Delivery, Complications, and Care

All SBA infants from the fetal intervention groups (OSBAR and FSBAR) were born by Cesarean section. All SBA infants from the matching NSBAR group were born by vaginal delivery. In the perspective of old literature regarding Cesarean Section vs vaginal delivery [22], we controlled for a potential delivery effect on dMUD prior to the study. In 13 age- and lesion-matched NSBAR pairs for the way of delivery, we observed no effect, which is in line with the current literature [10,23,24,25,26,27,28,29]. 

In each intervention group, we collected clinical obstetric and neonatal complication data, Table 3. All SBA infants from the three participating university centers (Bonn, Katowice, and Groningen) received multidisciplinary care according to international standards.

“OSBAR-NSBAR” matched pairs: After informed consent, we included 17 infants following open fetal intervention (OSBAR) from the Medical University of Silesia Poland, treated from 2007–2017. The fetal surgeons from this team are trained by the team of Adzick et al., according to the guidelines of the MOMS [4]. The Polish team has reported equivalent results as the MOMS trial [30,31]. We matched each child from the OSBAR group with a child from the NSBAR group of the same MMC level and comparable age. The NSBAR-treated children had been operated on by the Dutch UMCG team, performing and reporting surgical neonatal SBA procedures and neurological outcome data for decades [1,10], in line with international literature standards [32].

“FSBAR-NSBAR” matched pairs: For outcome comparison between “OSBAR- NSBAR” versus “FSBAR-NSBAR,” we included 13 age- and lesion-matched “FSBAR- NSBAR” pairs, which were previously assessed and reported according to the same methods and performed by the same neurologic investigators [10].

### 2.2. Methods

#### 2.2.1. Clinical Parameters

In two phases, we prospectively obtained the data from the fetally operated groups, according to the Groningen study protocol. The data from the FSBAR group were investigated and reported first [10]. For the present study comparison, we additionally obtained the data according to the same protocol in the FSBAR group. All data from the NSBAR control children were already obtained (by the same investigators) and stored in a clinical research database. After matching, this data were retrospectively included.

##### Determination of the Anatomic Level of the MMC

The prenatal level of the MMC was depicted at the upper border of the MMC, as determined by fetal -ultrasonography and confirmed by neonatal MRI, in all 3 groups. According to Sherrod et al., fetal -MRI and -ultrasonography are equally effective in determining the level of the MMC [33].

##### Shunt Dependency

For clinical comparison, we evaluated shunt dependency in age- and lesion-matched intervention groups: 1. OSBAR versus NSBAR, and 2. FSBAR versus NSBAR (historical data [10]). We compared shunt dependency between both groups of matched pairs. 

#### 2.2.2. Primary Outcome Parameters on Segmental Neurologic Function

##### Muscle Ultrasound Density (MUD)

In SBA, the assessment of MUD parameters is based on secondary muscle alterations after damaged neural innervation. These secondary alterations involve reduced muscle water content, fibrosis, fat deposition, and atrophy, causing increased reflection of the muscle ultrasound beam, and resulting in increased MUD values. MUD in muscle segments cranial to the MMC can be influenced by cerebral and spinal innervation cranial to the MMC, whereas MUD in muscle segments caudal to the MMC can be influenced by hampered innervation cranial to the MMC and by spinal pathology at the MMC (i.e., the neural tube defect). By comparison of the quantitative MUD value caudal to the MMC with the MUD value cranial to the MMC, the impact of the MMC upon the muscle condition caudal to the MMC can be derived (see Figure 1). After the fetal intervention, a lower dMUD value, in comparison with the NSBAR matched pair, would implicate less impact by the 2nd hit of damage at the MMC and thus more preserved segmental muscle integrity. In accordance with previous studies [10], we included muscle ultrasound registrations of biceps, quadriceps, and calf muscles (see also legends Figure 1). All measurements were obtained at standardized reference points under the same settings for muscle ultrasound gain, dynamic range, compression, and time-gain compensation [10]. For digital quantification, we stored five ultrasound images per muscle and determined MUD within a well-defined region of interest. MUD outcome is derived by excluding the highest and lowest values and by calculating the mean of the three remaining MUD values. In order to minimize variation and bias, all muscle ultrasound recordings and calculations were performed by the same investigators (RJV). All fetal OSBAR and FSBAR recordings were performed with the same portable ultrasound equipment (LOGIQ e; GE Health-care, Jiangsu, China). All NSBAR muscle ultrasound recordings were performed with fixed ultrasound equipment (LOGIQ 9; GE Healthcare). Each of the two matched groups of fetal versus neonatal intervention is thus influenced by the same factor of the ultrasound machine (OSBAR-NSBAR and FSBAR-NSBAR), allowing direct comparison between both matched groups. Portable and fixed muscle ultrasound machines are compatible with GE Healthcare LOGIQ systems. For outcome comparison, we computed convertible MUD values in accordance with the machine by MUD logiq 9 = 37.262 + 1.368 * MUD logiq e [r^2^ = 0.74] [10].

##### The Intra-Individual Difference in Muscle Ultrasound Density (dMUD)

We computed the intra-individual difference of muscle ultrasound density parameters by dMUD = (MUD-caudal to the MMC) minus (MUD-cranial to the MMC), representing the effect of the MMC on segmental muscle integrity (Figure 1). In all infants, the calf muscle (S1) represented the standard muscle for MUD assessment caudal to the MMC. In all infants with a thoracic level to the high lumbar lesion (≥L2-3), the biceps muscle represented the standard muscle for MUD assessment cranial to the MMC [dMUD = (MUD-calf muscle) − (MUD-biceps muscle)]. In all infants with a low lumbar or sacral level of the lesion (≤L3-4), the quadriceps muscle represented the standard muscle for MUD assessment cranial to the MMC [dMUD = (MUD-calf muscle) − (MUD-quadriceps muscle)]. See also Figure 1.

We compared dMUD between children from the OSBAR group versus outcomes in age and lesion-matched children from the NSBAR group. Finally, we associated these results with the comparative dMUD results between the FSBAR versus the age- and lesion-matched NSBAR group (historical data [10]). 

In this manner, we obtained comparative dMUD values between both age- and lesion-matched groups (1. OSBAR-NSBAR versus 2. FSBAR-NSBAR).

##### Segmental Sensory- and Motor- Assessment

We included standardized neurological outcome data obtained according to the previously described methods by the same pediatric neurologist (DAS [10]). Neurological examinations were videotaped and scored offline. Sensory levels were indicated by the cranial dermatome at which a pinprick still elicited an emotional response. Motor levels were indicated by the most cranial myotome involved in active motor behavior. In infants in whom neurological levels were different on the left and right sides, we took the calculated mean of the segmental levels from both legs. For statistical comparison between age- and lesion-matched pairs, we attributed numerical scores to each neurological level ranging from 0 to 8 (i.e., T12 = 0; L1 = 1; L2 = 2; L3 = 3; L4 = 4; L5 = 5; S1 = 6; S2 = 7; and no neurological dysfunction = 8). For outcome comparison of segmental sensory and motor function, we compared outcomes between both age- and lesion-matched groups (1. OSBAR-NSBAR versus 2. FSBAR-NSBAR).

##### Reflex Activity

For analysis of leg reflex activity, we examined knee-jerk (L2–4) and anal reflexes (S3–5). Knee-jerk reflexes were evoked in the supine position. We attributed a score of ‘2′ to visible reflexes in both legs, ‘1′ to a visible reflex in one leg, and ‘0′ to invisible reflexes. The anal reflex was evoked in the prone position and scored offline as present (visible sphincter contractions at both anal sides: ‘2′ points), weak (sphincter contractions at one side: ‘1′ point), or absent (no visible contractions: ‘0′ points).

For outcome comparison of reflex activity, we compared reflex outcomes between both age- and lesion-matched groups (1. OSBAR-NSBAR versus 2. FSBAR-NSBAR).

##### Primary and Secondary Outcome Measures

Primary outcomes for segmental neurologic function:

1. the influence of the MMC on muscle ultrasound density (MUD): MMC (dMUD = [MUD_caudal-to-the-MMC_] − [MUD_cranial-to-the-MMC_]). 2. segmental motor function, 3. segmental sensory function, 4. reflex activity.

#### 2.2.3. Secondary Outcomes for Ambulation

We assessed and compared the potential significance of the primary outcome (i.e., dMUD value in comparison with the matched NSBAR control, representing the altered 2nd hit of damage by the fetal intervention) for ambulation between both fetal treatment groups. We reasoned that the theoretically calculated likelihood of gained ambulation provides the information on the potential effect of the fetal intervention more directly than the actual follow-up results of ambulation since the latter parameter is also subject to the heterogeneity of clinical circumstances. We derived the theoretical likelihood of potentially “gained” ambulation from the segmental motor function gain [24]. The likelihood of theoretically “gained” ambulation can individually be assessed as [%prognosis to acquire ambulation in accordance with the assessed neurologic segmental motor function] minus [%prognosis to acquire ambulation in accordance with the fetal radiologic MMC level]. The %prognosis to acquire ambulation is characterized by 3 groups (I, II, and III, see below) following the natural disease course after neonatal MMC closure [34,35,36,37], in accordance with the Dutch SBA rehabilitation guidelines [38].

A. Group I: L4-S4: likely walkers (prognosis for outside walking 83–100%) [34,35,36]B. Group II: L3: potential walkers (prognosis for outside walking with long ortheses: 33–60%) [34,35,36]C. Group III: Th-L2: unlikely walkers (3–24% walkers; mostly wheelchair dependent) [34,37]

Following the fetal intervention, we interpreted a segmental motor function “gain” as functional when the neurologically assessed motor segment would relate with a better prognostic group than the fetal radiologic level of the MMC. This implicates a more favorable prognostic group assignment for potential ambulation after fetal intervention (i.e., from group III to group II or I or from group II to group I). We subsequently determined and compared the number of group transitions between fetal -OSBAR and FSBAR strategies.

#### 2.2.4. Tertiary Outcomes for Ambulation

Finally, we descriptively collected information on the actual functional ambulation of the children included. Functional (short and long-term) walking was characterized as the ability to walk during daily activities, either unsupported or with short ankle-foot orthoses. Children needing long orthoses or a stroller were excluded from this parameter. Short-term walking is determined by the possibility of initially walking (i.e., before school age). Long-term walking is determined by the possibility of persistently walking (i.e., until to date). The information was obtained from the parents (by e-mails and phone calls) and, when legally allowed, from patient files. Due to the subjective character of this parameter, we did not stratify for walking distance or walking quality. As this parameter is potentially influenced by multiple individual factors (other than the second hit of damage), we provide this outcome parameter in a descriptive way.

#### 2.2.5. Statistical Analysis

In the present descriptive study, we applied statistical analysis using SPSS, version 25.0 (SPSS Inc., Chicago, IL, USA). As MUD values and neurological parameters were not normally distributed (according to Q–Q plots and the Shapiro–Wilk test), we compared matched pairs by non-parametric Wilcoxon signed-rank test. Group comparisons (between FSBAR and OSBAR) were performed by the Mann-Whitney U test. The level of significance was *a =* 0.05.

## 3. Results

### 3.1. Clinical Data

Comparing patient inclusion in the fetoscopic versus the open fetal surgery group revealed statistically similar outcomes regarding 1. the segmental level of the cranial boundary of the MMC medial cranial level L3 vs. L4, respectively *(ns),* 2. the number of segments covered by the MMC median 2 (range 1-6 segments) vs median 2 (range 1–7 segments) (*ns*). The intervention-related complications are shown in Table 3 (OSBAR, FSBAR [10], and NSBAR). Fetal interventions (OSBAR and FSBAR) were both associated with more intervention-related complications than NSBAR. Comparing fetal interventions revealed more and also more severe intervention-related complications after the FSBAR (historical data [10]) than the OSBAR approach, see Table 3.

The prevalence of shunt dependency was lower after fetal than neonatal- intervention [OSBAR versus NSBAR (4/17 versus 14/17); FSBAR versus NSBAR (4/13 [10] versus 12/13), both *p* < 0.05]. The prevalence of shunt dependency did not significantly differ between OSBAR and FSBAR (4/17 versus 4/13); *p* = 0.742].

### 3.2. Segmental Neurologic Outcome Parameters

#### 3.2.1. Difference in the MUD (dMUD)

In children operated by OSBAR, age- and lesion-matched dMUD-values varied between −26 and 78 (median 21). Comparing dMUD between OSBAR (ca ≈ 21.0), NSBAR (ca ≈ 27.0), and FSBAR (ca ≈ 15.0) historical data [10]) revealed: 1. no significant differences between OSBAR versus NSBAR (*p* = 0.471). 2. significant differences between FSBAR versus NSBAR (historical data; *p* ˂ 0.05) [10], 3. no significant differences between OSBAR versus FSBAR (median difference 6; *p* = 0.744). For dMUD outcome comparison between fetal and neonatal intervention, see Figure 2.

#### 3.2.2. Sensory Segmental Function

Age- and lesion-matched comparison revealed more preserved segmental sensory function after fetal (OSBAR and FSBAR) than neonatal (NSBAR) intervention [median difference 1.75 dermatomes (range −2 to 5); *p* = 0.001]. We observed the strongest segmental gain in fetally operated children with MMC lesions at L3 in comparison with the other MMC lesions, see Figure 3. Comparing segmental sensory function between OSBAR versus NSBAR revealed more preserved leg sensory function in the OSBAR group [median 1.0 dermatome (range −2 to 5); *p* = 0.019], see Figure 4a. Comparing segmental sensory function between FSBAR versus NSBAR revealed more preserved leg sensory function in the FSBAR group [median 2 dermatomes (range 1.5 to 5) [10]. Segmental sensory outcome comparison between OSBAR versus FSBAR did not reveal a significant difference (*p* = 0.213). For group comparison, see Figure 5a.

#### 3.2.3. Motor Segmental Function

Segmental motor function was more preserved after fetal (OSBAR and FSBAR) than neonatal (NSBAR) intervention [median difference of 1 myotome (range −2.5 to 6); *p* = 0.008]. We observed the strongest segmental motor function gain in fetally operated children with MMC lesions at L3 when compared with other MMC lesions [MMC at L3-L4: 1.5 (−1 to 4) versus [MMC at Th12-L2 and L5-S1: 0.5 (−1.5 to 3.5) segments; *p* = 0.026; median (range)]; see Figure 3 and Figure 6. In the fetally operated children, there was no significant association between fetal-intervention-related segmental motor function gain and postnatal age of the investigated child *(r* = −0.132; *p* = 0.329). Comparing leg motor function between OSBAR and NSBAR did not reveal a difference (median difference of 0 myotomes (range −2.5 to 6); *p* = 0.326). See Figure 4b. Comparing leg motor function between FSBAR and NSBAR revealed more preserved leg motor function in the first group [median 2 myotomes (range 0.5 to 4) [10]. Comparing leg motor function between OSBAR and FSBAR [10] did not reveal a significant difference; *p* = 0.086. For group comparison, see Figure 5b.

#### 3.2.4. Reflex Activity

Comparing leg reflexes between fetal (OSBAR and FSBAR) versus neonatal (NSBAR) intervention groups revealed more preserved reflexes in the fetal-intervention group (50⁄60 versus 21⁄60 points, respectively; *p* = 0.001). The presence of preserved leg reflexes did not significantly differ between OSBAR and FSBAR intervention groups (27/34 versus 22/26, respectively; *p* = 0.432). Comparing anal reflexes between fetal (OSBAR and FSBAR) versus neonatal (NSBAR) intervention revealed more preserved reflexes in the fetal group (14/60 versus 2⁄60 points, respectively, *p* = 0.039). Comparing preserved anal reflexes between OSBAR versus FSBAR intervention groups did not reveal a significant difference (8/34 versus 11/26, respectively; *p* = 0.742 *(ns)*).

### 3.3. Secondary Outcome: Predictions for “Gained” Ambulation

Comparing the fetal (OSBAR and FSBAR) and neonatal (NSBAR) intervention groups revealed a higher percentage of potential walkers in the fetal intervention group (segmental motor function at- or caudal to L3; *p* = 0.008). Comparing the functional significance of segmental motor function gain for the likelihood of acquiring ambulation did not reveal significant results between OSBAR and NSBAR [≤L3 or caudal; *p* = 0.157]. Comparing the functional significance of segmental motor function gain for the likelihood to acquire ambulation revealed significantly more potential walkers in FSBAR than NSBAR (i.e., segmental motor function at- or caudal to L3; *p* = 0.023). In 20% of the fetally operated children, we anticipated a transition to a better prognostic group because of segmental motor function gain (OSBAR 2/17 (12.0%); FSBAR: 4/13 (30.0%). Comparing prognostic group transitions between OSBAR and FSBAR interventions did not reveal a significant difference.

### 3.4. Tertiary Outcome: Actually Reported Functional Ambulation

We collected descriptive information on the reported functional ambulation (both short and actual long-term) in 17, 9, and 17 children (NSBAR, FSBAR, and OSBAR, respectively). In the reported children from the NSBAR, FSBAR, and OSBAR groups, the median, upper level of the MMC was at L4 (range Th12-S1 (NSBAR)), L3 (range Th12-L4 (FSBAR)) and L4 (range Th12-S1 (OSBAR)). In the short term, functional walking was reported in respectively 58%, 89%, and 88% (NSBAR, FSBAR, and OSBAR groups, respectively). In the long term, functional walking was reported in respectively 41%, 78%, and 71% (NSBAR, FSBAR, and OSBAR groups, respectively).

## 4. Discussion

In SBA, we investigated and compared the results after two fetal intervention strategies (OSBAR and FSBAR) in association with the standard neonatal SBA repair technique (NSBAR). To the best of our knowledge, this is the first comparative study between three age- and lesion-matched intervention strategies. In comparison with neonatal intervention (NSBAR), fetal intervention (OSBAR and FSBAR) revealed fewer shunt implantations, smaller dMUD values (reflecting less impact by the 2nd hit of damage), more preserved segmental neurologic (sensory and motor) function and more likelihood to acquire ambulation in ≈20% of the fetally operated children. Fetal intervention at L3 was associated with more segmental motor function gain and, thus a higher likelihood to “gain” ambulation than children with other lesions. Neurological outcome comparison between OSBAR versus FSBAR revealed no significant differences. However, the age- and lesion-matched FSBAR group tended to reveal more favourable outcomes in neurologic “gain” than the age- and lesion-matched OSBAR group. Conversely, the age- and lesion-matched FSBAR group revealed more and also more severe intervention-related complications than the present age- and lesion-matched OSBAR group [10]. However, it is recently reported that the FSBAR technique has shown a learning curve, which may result in similar complication risks for FSBAR, OSBAR, and NSBAR [8]. Especially in the light of possibilities for early detection of NTDs [39], these findings may have implications for the neurologic counselling of patients in the future. However, before clinical implementation, we recommend awaiting the results of a well-powered, large randomized controlled trial in the future.

In former studies, we have shown that the muscle ultrasound technique can be applied to determine the intra-individual impact of the MMC on neuromuscular integrity [10,15,21]. Our data show that histologic muscle integrity caudal to the MMC is better preserved after the fetal intervention, reflecting the neuroprotective effect of the intervention. These outcomes can theoretically be attributed to a neuroprotective effect against inflammation [17], neurotoxicity [2,20], and/or vascular damage [2]. In addition to a neuroprotective effect on muscle integrity, fetal intervention groups also revealed better segmental motor- and sensory- outcomes and a higher likelihood of acquiring ambulation in ≈20% of the fetally operated children. As fetal therapeutic “gain” could theoretically decrease over time (especially when the neural placode tissue deteriorates [2], when the child gains in weight [40], or when Chiari malformations and/or tethered cord symptoms evolve [41]), we subsequently evaluated the longevity of these results. However, we did not observe an age-related loss of gained motor function over time. These results are in accordance with previous SBA reports on fetal intervention, both in children up to 3 years of age [8] and in children of 5 to 10 years of age [42], reporting no age-related deterioration.

In operated fetuses at L3 levels, we observed the most segmental gain. Although the underlying reason is unknown, it is tempting to speculate that a median gain of 1 or 2 segments in children with L3-4 MMC levels could allow propagation to secondary neurulation. This secondary neurulation involves a different process by local cell division and migration of neural cells within the mesodermal tissue of the conus area [43,44]. This could theoretically explain why fetally operated children also revealed significantly more preserved anal reflexes, which are innervated by the conus area after secondary neurulation. From a theoretical rehabilitation perspective on ambulation, fetuses operated at L3 levels can shift from the prognostic group of “potential” walkers to the group of “likely” walkers after gaining 1–2 segments [34,35], resulting in a ≈ 40% increased likelihood of acquiring ambulation (i.e., from 33–60% to 83–100%) [34,35]. This is contrasted by SBA children with thoracic to high lumbar lesions, which would theoretically need a motor function gain of at least three spinal segments to shift from the group of “unlikely” walkers to “potential” walkers, with ≈ a 25% increased likelihood to acquire ambulation (i.e., from 3–24% to 33–60%) [34,35]. Whereas for fetuses operated at L5-S1 levels, the likelihood of acquiring walking by segmental gain remains the same (i.e., a stable “likely” prognosis for outside walking is already present before gain: 83–100%) [34,35].

Comparison between fetal intervention groups (OSBAR and FSBAR techniques) revealed no significant differences in dMUD outcomes, although the FSBAR-treated group tended to reveal more statistically convincing results when compared with NSBAR controls. On the one hand, this could theoretically be attributed to the minimal invasiveness of the endoscopic procedure (FSBAR), resulting in better tissue preservation [45]. However, on the other hand, it may also be argued that the median lesion level of the FSBAR group appeared at a more favorable level for segmental gain than the OSBAR group (i.e., L3 and L4, respectively). As fetally operated children with MMC at L3 tended to have the most favorable segmental gain when compared with controls, one cannot fully exclude that this could have contributed to the results. A future randomized controlled trial with stratification for MMC levels may hopefully elucidate this point to a further extent. Analogous to dMUD results, we also observed no significant differences comparing segmental neurological outcome parameters between OSBAR and FSBAR intervention groups, although the FSBAR-intervention group tended to reveal more convincing results when compared with NSBAR controls. In contrast, the iatrogenic risk for complications after the fetal intervention was less in the OSBAR than in the previously assessed FSBAR [10] intervention group. As shown by our data (Table 3), fetoscopy resulted in more premature delivery than the open approach [46]. In the literature, it is indicated that fetoscopy, in general, presents a higher risk of PROM and premature labor as compared to the open approach, which has been attributed to more difficulties in achieving a triple-layer waterproof closure [46,47]. Furthermore, it has been indicated that the open fetal surgical approach is often performed at an earlier gestational age than the fetoscopic approach [46,47], which could reduce the shunt rate [46]. However, in the present study, we did not observe a significant difference in shunt rates between both fetal techniques. Interestingly, a recent report on the FSBAR technique indicated that the complication risk of the FSBAR technique has now been improved, resulting in similar mortality and morbidity (including prematurity) as in OSBAR interventions [8]. Altogether, growing experience and improvement of fetal techniques, this may implicate that the outcome of SBA can be improved [48], especially when stratified results become available so that parents can select the technique that suits the individual characteristics and goals best. However, before such clinical implication becomes feasible, we should await the results of a larger randomized controlled trial.

In the present descriptive study, we are aware of sample size limitations related to the rarity of the three cohorts collected over a time span of almost 10 years. First, although the dMUD value of open fetal surgery did not reach the level of significance in comparison with controls, this does not automatically imply that open fetal surgery is not beneficial for tissue preservation. Since the dMUD value was significantly lower for the fetal endoscopic group versus controls and since there was no statistical difference in dMUD values comparing endoscopic versus open fetal surgery in relation with age- and lesion-matched controls, preserved muscle tissue integrity by open fetal surgery is implicated. Furthermore, in the perspective of significantly preserved segmental sensory- and muscle- function after fetal (open and endoscopic) versus neonatal operation, preserved segmental neurologic function after the fetal (open and endoscopic) intervention is implicated. Hopefully, future randomized controlled trials will quantify this to a further extent. Second, as the investigated parameters were all targeted at the research question exploring the potential existence of segmental “gain” by fetal surgery (instead of ad randomly chosen parameters), testing with multiple comparisons was not applied to the current approach. However, comparing the investigated parameters between fetal (FSBAR and OSBAR) versus neonatal (NSBAR) intervention would still reveal significant results *(p* < 0.05) for all segmental neurological parameters (i.e., sensory function, motor function, and preserved leg reflexes) when corrected testing for multiple comparisons would be applicable. The same also applies to significantly reduced shunt-dependency in the fetal versus neonatal operation groups (either with or without testing for multiple comparisons). Third, we included FSBAR data from a previously described cohort [10], implicating that the latest effects on the complication risks in FSBAR could not be taken along [8]. Fourth, we are aware of old literature indicating that a caesarean section prior to delivery could be beneficial [22]. However, the applied methodology was disputed [25,26] and results could not be replicated by subsequent study groups, including a study on an even larger patient cohort and also our own dMUD data quantifying the “2nd hit of damage” between 13 matched pairs of children born after vaginal delivery versus caesarean section [10,23,24,25,26,27,28,29]. Furthermore, we also checked our NSBAR outcomes by comparing the predicted motor segments from the anatomical MMC level versus the actual level of motor function, revealing a median difference of 0 segments (range −2.0 to + 2.0 segments). Finally, as both fetal intervention groups (OSBAR and FSBAR) are *all* matched with vaginally delivered NSBAR-intervention children, comparative fetal intervention results cannot be attributed to the delivery mode.

Fifth, in the absence of large study numbers and randomized assignment to the fetal treatment groups, one needs to control for the similarity of the included MMC lesions per group. This was partly overcome by the fact that we do not provide direct outcome comparisons between both fetal techniques, but the age- and lesion-matched results in comparison with NSBAR controls, instead. Furthermore, the included MMC lesions per fetal group were statistically similar *(ns)* regarding 1- the segmental level of the MMC and 2. the number of segments over which the MMC extended. However, the median lesion level of the FSBAR group was still one segment higher than that of the OSBAR group (i.e., L3 versus L4, respectively). Hopefully, future stratification in a large randomized controlled trial may elucidate this point to a further extent. The sixth limitation is that we cannot evaluate potential fetal intervention group differences for fetal or neonatal demise. However, in a recent study, it was suggested that mortality after FSBAR, OSBAR, and NSBAR might have become comparable (5.6%, 4%, and 3%) [8]. In the future, a randomized controlled trial may hopefully elucidate this critical issue to a further extent. Finally, we are aware that local clinical policies may differ between different centres. However, all included infants were treated by experienced and well-recognized European centres and assessed by the same experienced paediatric neurologists (RV; DAS) who are not engaged in fetal intervention strategies.

In conclusion: In SBA, fetal (open and fetoscopic) interventions are associated with positive neuroprotective effects. In ≈20% of the fetally operated children, the neuroprotective effects resulted in an increased likelihood of gaining ambulation. Fetuses operated at L3 levels revealed the most likely to acquire a higher likelihood of ambulation due to favorable intervention-related segmental “gain.” Comparing the OSBAR versus FSBAR interventions for neurologic segmental gain revealed no significant differences (*ns*), although the FSBAR tended to reveal more favourable results when compared with NSBAR controls, as previously indicated, at the risk of more frequent and more severe complications. However, in the perspective of a recently reported learning curve, this may have been improved in the meantime [8]. For further substantiation, one may thus await the results from a randomized controlled trial. We hope that the present and future information will contribute to the counselling of patients expecting a child with an open neural tube defect.

## Figures and Tables

**Figure 1 diagnostics-13-00251-f001:**
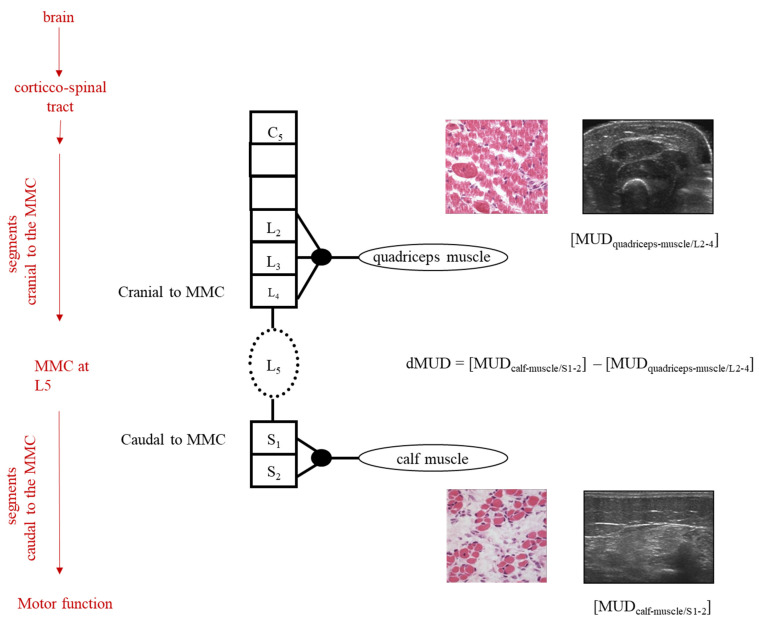
Schematic illustration of dMUD calculation in case of an MMC at spinal segment L5. On the left, the red arrows represent the influences on the neurological examination of the child. The black column represents the segmental innervation of the spinal cord, cranial-, and caudal to the MMC. In the case of an MMC at L5, the quadriceps muscle (innervation L2-4) is innervated by motor neurons originating cranial to the MMC and the calf muscle by motor neurons originating caudal to the MMC. By intra-individual subtraction of the MUD value caudal to the MMC minus the MUD value cranial to the MMC, the dMUD-outcome reflects the effect of the 2nd hit of damage on the muscle caudal to the MMC (i.e., the calf muscle). In children with an MMC at higher levels (i.e., at thoracic and/or high lumbar levels), dMUD values can be similarly calculated using the Biceps muscle (innervation C5) instead of the quadriceps muscle.

**Figure 2 diagnostics-13-00251-f002:**
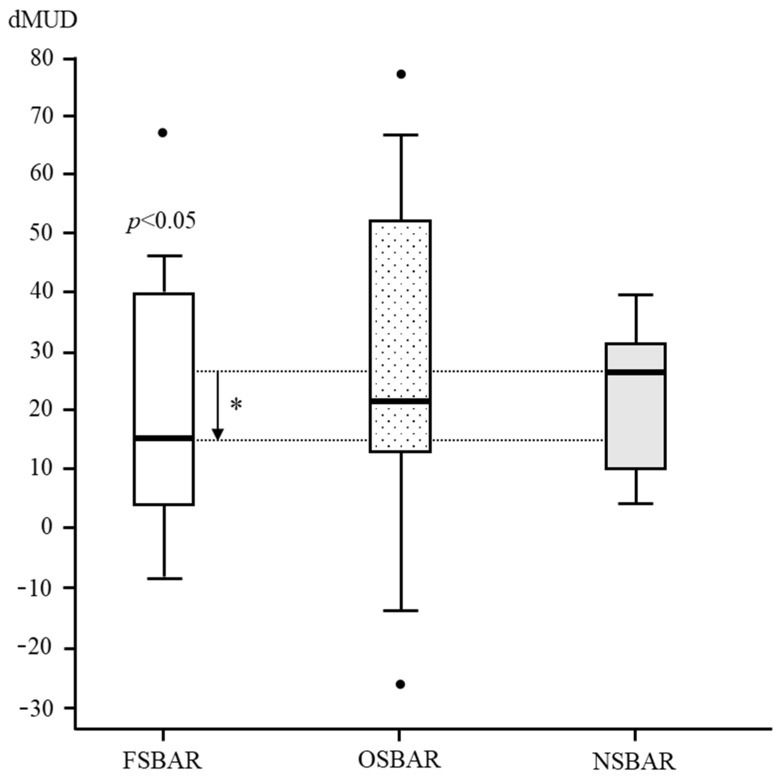
dMUD comparison between FSBAR, OSBAR, and NSBAR interventions. dMUD comparison between the three treatment groups: 1. fetal–endoscopic (FSBAR), 2. open fetal (OSBAR), and 3. neonatal (NSBAR) intervention. The *x*-axis indicates FSBAR, OSBAR, and NSBAR, intervention groups. The *y*-axis indicates intraindividual dMUD values per cohort. dMUD values reflect the impact of the MMC on neuromuscular integrity. A lower dMUD value implicates less impact by the MMC and thus better preserved segmental muscle integrity caudal to the MMC (for a methodologic explanation, see Figure 1). Quantitative dMUD values were significantly lower (better) in the FSBAR than the NSBAR group, *p* < 0.05. Differences between OSBAR versus NSBAR and between FSBAR versus OSBAR were not statistically significant. Box plots mark the first and third quartiles; whiskers represent data points 1.5 times the interquartile range below and above the first and third quartiles. dMUD= (MUD-caudal to the MMC) minus (MUD-cranial to the MMC). Abbreviations: MMC = myelomeningocele; FSBAR = fetal intervention technique by the endoscopic approach; OSBAR = fetal intervention technique by the open approach; NSBAR = neonatal MMC closure; MUD = muscle ultrasound density; dMUD = the difference in muscle ultrasound density over the MMC (*= *p* ˂ 0.05; Dots represent outliers).

**Figure 3 diagnostics-13-00251-f003:**
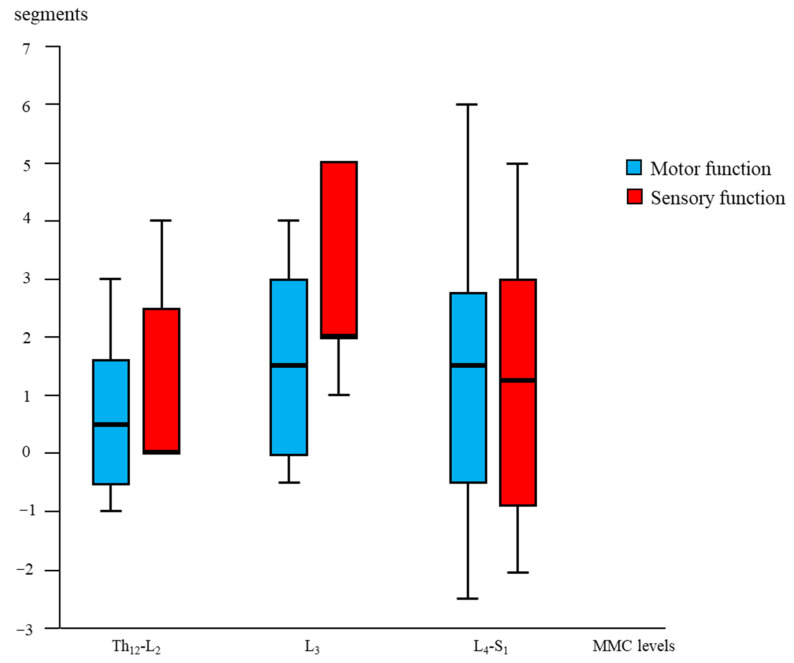
Segmental “gain” after fetal (OSBAR and FSBAR) intervention. Combined OSBAR- and FSBAR- data are provided from the perspective of the lesion-matched NSBAR group. The *x*-axis indicates the segmental MMC levels of the included age- and lesion-matched children. The *y*-axis indicates the segmental difference between fetal versus neonatal intervention. Red bars indicate sensory function, and blue bars indicate motor function. Positive values indicate the number of gained segments after fetal versus neonatal intervention. Box plots mark the first and third quartiles; whiskers represent data points 1.5 times the interquartile range below and above the first and third quartiles. Fetal intervention at L3-L4 lesions seemed most favorable. Abbreviations: MMC = myelomeningocele; fSBA = the total group of fetal interventions (OSBAR + FSBAR); NSBAR-total = the total group of neonatal interventions (age- and lesion-matched with fSBA).

**Figure 4 diagnostics-13-00251-f004:**
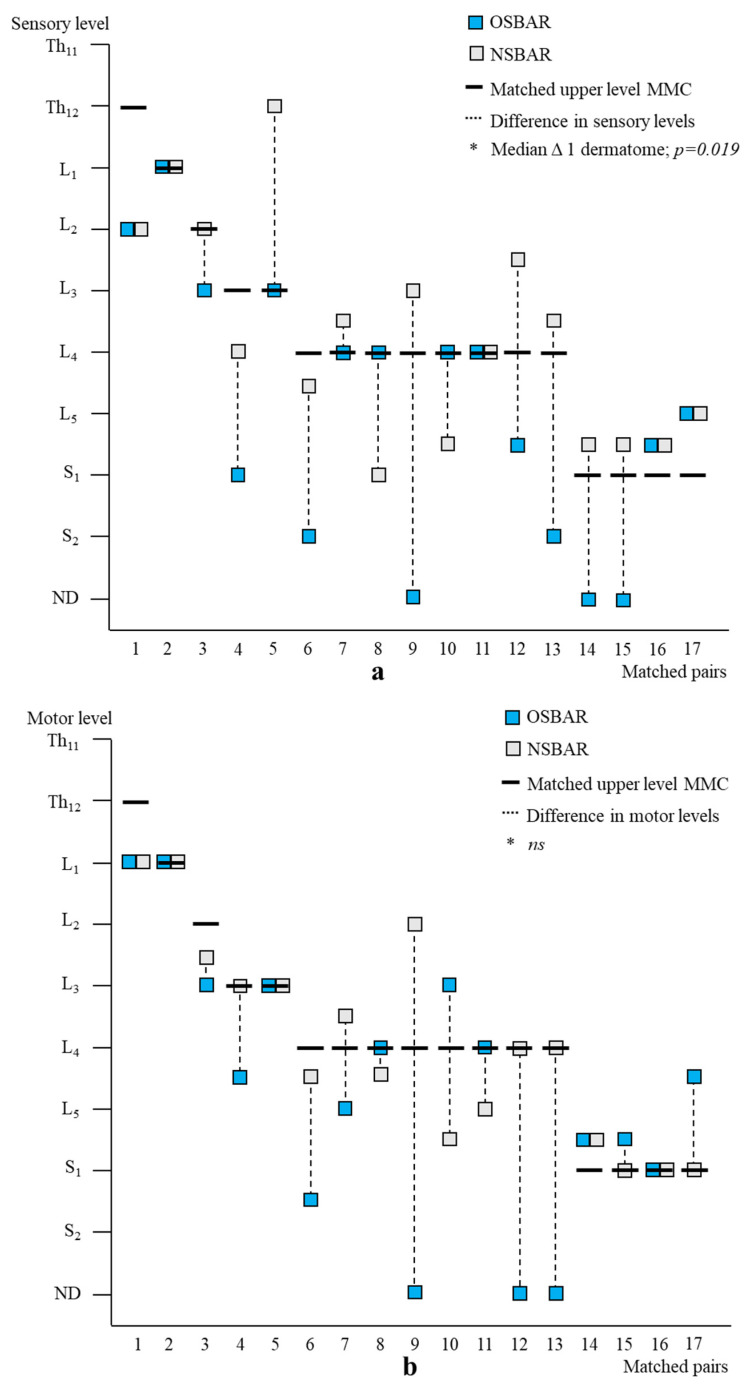
Segmental neurologic outcome comparison between OSBAR and NSBAR. (**a**): Segmental sensory function of age- and lesion-matched pairs after OSBAR and NSBAR interventions. The *x*-axis indicates the matched pairs. The *y*-axis indicates the segmental sensory function. Sensory function was better preserved in OSBAR than in NSBAR. * Median difference 1 dermatome; *p* = 0.019. (**b**): Segmental motor function of age- and lesion-matched pairs after OSBAR and NSBAR interventions. The *x*-axis indicates the matched pairs. The *y*-axis indicates the segmental motor function. The motor function did not significantly differ between OSBAR and NSBAR; *ns*= not significant. Abbreviations: ND = no deficit. MMC = myelomeningocele; OSBAR = fetal intervention technique by the open approach; NSBAR = neonatal MMC closure.

**Figure 5 diagnostics-13-00251-f005:**
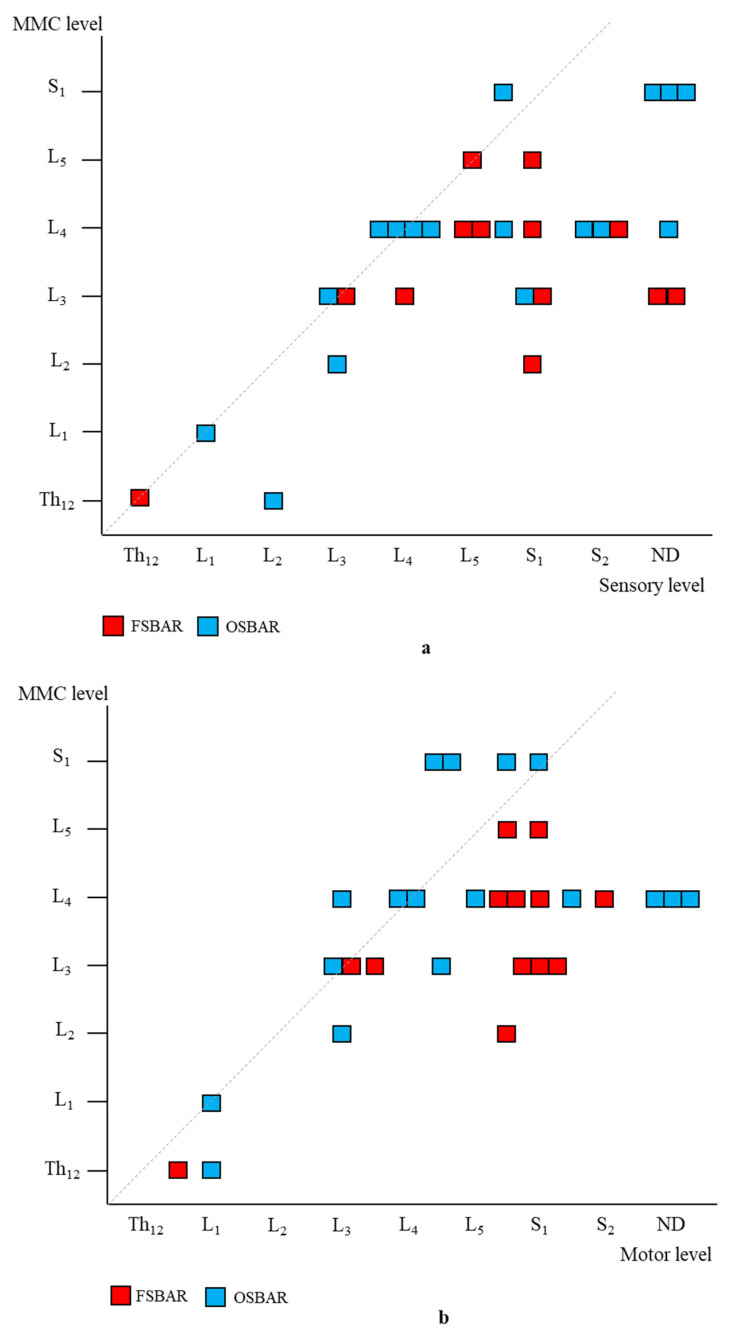
Segmental neurologic comparison between OSBAR and FSBAR. (**a**): Radiologic level of the fetal MMC versus segmental sensory function in OSBAR and FSBAR. (**b**): Radiologic level of the fetal MMC versus segmental motor function in OSBAR and FSBAR. The *x*-axis indicates the segmental neurologic function (sensory function, (**a**), and motor function, (**b**)). The *y*-axis indicates the segmental level of the MMC. FSBAR cases are indicated by red squares, and OSBAR cases are indicated by blue squares. The dotted lines represent the expected outcomes when the segmental neurological (sensory, motor) function corresponds with the MMC level. FSBAR cases are indicated by red squares, and OSBAR cases are indicated by blue squares. Segmental sensory and motor function gain did not statistically differ between OSBAR and FSBAR interventions, but results after FSBAR tended to be better. Abbreviations: MMC = myelomeningocele; OSBAR = fetal intervention technique by the open approach; NSBAR = neonatal MMC closure.

**Figure 6 diagnostics-13-00251-f006:**
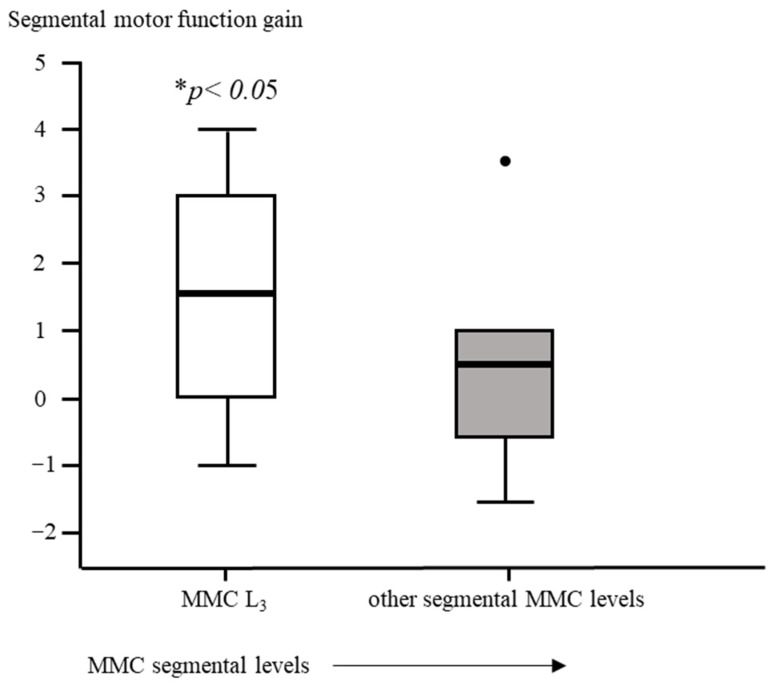
Intervention-related difference in segmental motor function between fetally and neonatally operated children subdivided per MMC-level. The x-axis represents a subdivision of MMC levels of L3-L4 or other. The y-axis represents the difference in segmental motor levels between fetal and neonatal MMC operations. Segmental motor function gain is determined by the difference in segmental motor function level between age- and lesion-matched pairs of fetal versus neonatal operations. Fetally operated children represent OSBAR and FSBAR intervention groups together. Fetally operated children with an MMC at L3-L4 revealed significantly more segmental motor function gain (* *p* < 0.05) than other lesions. Box plots mark the first and third quartiles; whiskers represent data points 1.5 times the interquartile range below and above the first and third quartiles. Dots represent outliers. MMC = myelomeningocele.

**Table 1 diagnostics-13-00251-t001:** Age- and MMC matched pairs: a. OSBAR versus b. NSBAR.

Pair		Matched upper Level MMC	Age at Assesssment
1	a	Th_12_	7 m
	b		1 y
2	a	L_1_	6 m
	b		0 m
3	a	L_2_	1 y
	b		1 y
4	a	L_3_	6 m
	b		7 m
5	a	L_3_	4 m
	b		4 m
6	a	L_4_	1.5 y
	b		4 y
7	a	L_4_	2 y
	b		1.5 y
8	a	L_4_	8 y
	b		5 y
9	a	L_4_	11 y
	b		10 y
10	a	L_4_	6 y
	b		5 y
11	a	L_4_	2 y
	b		3 y
12	a	L_4_	11 m
	b		5 m
13	a	L_4_	10 m
	b		8 m
14	a	S_1_	2 y
	b		1.5 y
15	a	S_1_	1 y
	b		1 y
16	a	S_1_	5 y
	b		4 y
17	a	S_1_	2 y
	b		1 y

Legend: a = fetally operated; b = neonatally operated; MMC = myelomeningocele; Th = thoracal; L = lumbar; y = year(s); m = months.

**Table 2 diagnostics-13-00251-t002:** Matched pairs for MMC level and postnatal age.

Matched Pairs	FSBAR vs. NSBAR	OSBAR vs. NSBAR
Number Total	13 per group 2 × 13	17 per group 2 × 17
MMC level Median	Th12-L5 L3	Th12-S1 L4
Age Median	0–5 year 1 year	0–11 year 2 years

Legend: median and range of MMC levels and age of 2 matched pairs. MMC = myelomeningocele; OSBAR = fetal intervention technique by the open approach; FSBAR = fetal intervention technique by the endoscopic approach; NSBAR = neonatal MMC closure.

**Table 3 diagnostics-13-00251-t003:** Clinical Data.

	Significant Maternal Morbidity	Fetal or Neonatal Demise	Mean GA at Delivery in wks	Oligo- Hydramnios	Neonatal Infection	Respiratory Problems	Other
FSBAR * group N = 17 2003-09	N = 4 (24%)	N = 6 (35%)	29.0	N = 13 (77%)	N = 6 (35%)	N = 16 (92%)	Asphyxia N = 2 Endocr N = 2
OSBAR Group N = 13 2011-18	N = 0	N = 0	34.0	N = 1 (8%)	N = 4 (30%)	N = 3 (23%)	Femur # N = 1
NSBAR Group N = 25 2003-18	n.a.	N = 1 (4%)	38.0	N = 1 (4%)	N = 1 (4%)	N = 1 (4%)	Endocr N = 1

Legends: maternal morbidity = haemorrhage requiring RBC transfusion; pulmonary oedema, placental abruption, uterus dehiscence; FSBAR= Fetal Spina bifida aperta repair; OSBAR= Open fetal Spina bifida repair; GA= Gestational age at delivery; n.a. = not applicable; wks = weeks; endocr= endocrine; # = fracture; * = historical data [10].

## Data Availability

The data that support the findings of this study are available from the corresponding author, DAS, upon reasonable request.

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
