# Peer review of "Neurologic Outcome Comparison between Fetal Open-, Endoscopic- and Neonatal-Intervention Techniques in Spina Bifida Aperta"

_diagnostics, 2023, doi:10.3390/diagnostics13020251_

Round 1
Reviewer 1 Report
Meningomyelocele (MMC) is one of the most severe human malformations with significant impact on mobility and quality of life. If postnatal survival can be expected for a prolonged period, the current standard treatment in most centers is early postnatal closure of the MMC to avoid meningitis and further neurological damage. However, because deterioration of fetal leg movements has long been known to occur in the intrauterine period, intrauterine repair of the MMC has been introduced in specialized centers for several decades, initially with an open approach to the fetus and most recently with an endoscopic intrauterine approach. However, the highly specialized nature of the procedure, fetal mortality, prematurity, and infant and maternal morbidity, as well as its unclear impact on subsequent function and quality of life, have prevented its widespread acceptance and dissemination. The open surgical procedure was studied more than a decade ago in a very large, well-tested, randomized trial in the United States. It showed a significant reduction in the number of children who required a shunt to treat hydrocephalus, as well as a modest reduction in the degree of neurologic damage and an improvement in mobility by school age. To make the decision to refer a pregnant mother with an MMC fetus for such a procedure, both the referring physicians and the affected parents need the most objective information possible about the risks and prognosis for their child, which depend not only on the surgical method but, more importantly, on the extent of the malformation, i.e., the level of lesion at the spine and spinal cord.
In an attempt to gain more and objective information on this important topic, the authors studied a group of children with open intrauterine surgery compared with a postnatally operated group, individually matched for age and extent of MMC lesion (N=17 each). And they compared these results with their own historical data from a group of children operated on endoscopically in utero with a matched control group (N=13, published 10 years ago). The study methods were identical in the historical groups and in this study. In brief, both intrauterine methods resulted in fewer shunt placements, improvement in neurologic findings by a median of 2 spinal segments, and an estimated better mobility prognosis for 20% of children.
These data are an interesting contribution to ongoing discussions, but because of the small number of patients included and a number of major weaknesses (some of which are discussed by the authors themselves), the results cannot be generalized to the population and thus do not provide the objective data sought. The manuscript needs substantial improvements. To name a few:
1) Much reference is made in the manuscript to supplementary data and to the historical manuscript; however, for better understanding, the main methods and facts must be included in the manuscript itself.
2) The 2 tables must appear in the manuscript.
3) While the selection of postnatally operated infants is clearly stated (matched selection from the routinely operated Groningen group), the indication for intrauterine surgery in Poland and Bonn and the selection into the study for the open and endoscopic group is completely unclear. A strong selection bias in favor of fetuses with better prognosis must be assumed.
4) How was the anatomical MMC level determined prenatally? Ultrasound? MRI? Was this the same in all children?
5) Which of the data were collected retrospectively from the records and which prospectively according to an established protocol?
6) The intrauterine operated children were all delivered by cesarean section, but the postnatal operated children were delivered vaginally. There are some old retrospective and open data from Canada (but based on hundreds of cases) suggesting that cesarean section before the onset of uterine contractions spares 2 neurologic levels. The authors argue that this aspect is not important for their comparison between open and endoscopic surgery - but it is clearly important for their comparison with postnatally operated cases. And I strongly doubt that their own comparison of only 13 pairs of vaginally delivered vs. caesarean-delivered infants, which they cite in their historical publication, is valid statistical evidence that there is no difference in outcome.
7) Regarding the dMUD method, it is difficult to understand why the gluteus maximus is used as a non-affected reference in cases with low lumbar spine and sacral lesion, since the innervation of the gluteus comes from S1 (and the gluteus medius from L5), so both are affected in many of these cases.
8) As the detailed results in Fig. 3 show, the variability of anatomic lesions and functional segments is extremely high in all treatment groups. This is not apparent to the reader in the statistical values or in the boxplots. I therefore recommend that all data be presented in the form of Figure 3 and, in addition, that the Supplementum lists the data for all patients individually.
9) Given the small number of cases and the probable selection bias, the statistical analysis can only be descriptive and not confirmatory, as would be possible in a prospective and well-powered study. This should be clearly mentioned.
10) Have the statistical tests been corrected for multiple comparisons? For example, with the Kruskal-Wallis test?
11) The attempt to calculate a "gain in walking ability" on the basis of motor levels sorted into 3 groups is not convincing. It does not take into account the large variability, so that in individual cases the 20% improvement in walking ability says nothing about the actual walking ability. And is "walking in long orthoses" really a type of mobility a treatment aim that can be used on the playground or while shopping with the family?
12) The number of intrauterine and postnatal complications in the endoscopically operated group is extremely high and hardly acceptable. In reference (8), the same group of surgeons, after moving from Bonn to Giessen, reported a much larger number of endoscopically operated cases, with much fewer complications and no intrauterine mortality. This is only briefly mentioned in the manuscript, but is a very important aspect of the Bonn data that should be discussed very clearly.
13) The authors of reference (8) followed their patients until 30 months of age and can therefore present real data on ambulation. This should also be possible in the present manuscript, as the youngest patients underwent surgery in 2018 and it should be easy to obtain this information (and the patients from the endoscopic group are still in their teens).
14) It could be a very important finding that the children with an L3 level (anatomically? functionally?) benefited the most from the surgery. This could be a very important information and a selection criterion for surgery if it could be validated in a prospective and well-powered study.
15) It is not clear whether by "segmental gain" the authors mean the difference between anatomic and functional level or the difference between functional level in intrauterinely and postnatally operated infants.
Author Response
Reviewer 1: Introduction
Introduction: Meningomyelocele (MMC) is one of the most severe human malformations with significant impact on mobility and quality of life. If postnatal survival can be expected for a prolonged period, the current standard treatment in most centers is early postnatal closure of the MMC to avoid meningitis and further neurological damage. However, because deterioration of fetal leg movements has long been known to occur in the intrauterine period, intrauterine repair of the MMC has been introduced in specialized centers for several decades, initially with an open approach to the fetus and most recently with an endoscopic intrauterine approach. However, the highly specialized nature of the procedure, fetal mortality, prematurity, and infant and maternal morbidity, as well as its unclear impact on subsequent function and quality of life, have prevented its widespread acceptance and dissemination. The open surgical procedure was studied more than a decade ago in a very large, well-tested, randomized trial in the United States. It showed a significant reduction in the number of children who required a shunt to treat hydrocephalus, as well as a modest reduction in the degree of neurologic damage and an improvement in mobility by school age. To make the decision to refer a pregnant mother with an MMC fetus for such a procedure, both the referring physicians and the affected parents need the most objective information possible about the risks and prognosis for their child, which depend not only on the surgical method but, more importantly, on the extent of the malformation, i.e., the level of lesion at the spine and spinal cord.
In an attempt to gain more and objective information on this important topic, the authors studied a group of children with open intrauterine surgery compared with a postnatally operated group, individually matched for age and extent of MMC lesion (N=17 each). And they compared these results with their own historical data from a group of children operated on endoscopically in utero with a matched control group (N=13, published 10 years ago). The study methods were identical in the historical groups and in this study. In brief, both intrauterine methods resulted in fewer shunt placements, improvement in neurologic findings by a median of 2 spinal segments, and an estimated better mobility prognosis for 20% of children.
These data are an interesting contribution to ongoing discussions, but because of the small number of patients included and a number of major weaknesses (some of which are discussed by the authors themselves), the results cannot be generalized to the population and thus do not provide the objective data sought.
Response to the reviewer: We thank the reviewer for the provided recapitulation of the text and the remarks. We will firstly reflect on these points, before proceeding with the "questions and answers" – section.
In the previously well- performed and -communicated studies including the MOMS and endoscopically performed trials, resultant neurological "gain" was derived by associating the expected neurological outcomes based on the morphological upper level of the MMC with the final results of the actual neurologic examination. However, as indicated by the reviewer, no fetus with spina bifida (SBA) is the same as the other (regarding complications, cerebral malformations, effects by HC and CSF drainage etc.), resulting in heterogeneous effects on the final neurological examination. Since heterogeneity is the hallmark of SBA, a larger study sample (i.e., a better estimation of the mean) cannot prevent hampered generalizability to the individual patient at the outpatient clinic. This accounts for all studies on SBA.
In the present version of the manuscript, we now added this information to the introduction-section,
page 2, lines 58-64 and lines 67-69.
However, in the present study, we applied the muscle ultrasound technique, which quantifies the intra-individual difference in muscle ultrasound density of muscle segments innervated by spinal segments caudal vs cranial to the MMC. This parameter can thus quantify the individual effect by the MMC on muscle integrity. Comparison of this quantitative outcome parameter (i.e., dMUD) between different fetal and neonatal treatment groups can thus expose the severity of the 2nd hit of damage per intervention group. By creating age- and lesion matched groups with neonatally operated controls, comparison on the 2nd hit of damage becomes possible between fetal treatment strategies.
This is so, because the general, interindividual differences between cranial- and spinal- morphology & complication-related effects” are neutralized by computing the intra-individual difference MUD difference over the MMC in each child. Application of this technique thus diminishes the amount of heterogeneous, potentially confounding effects, when compared to the other studies. Until now, such objective, quantitative comparisons representing the 2nd hit of damage between age- and lesion- matched pairs of both FSBAR- and OSBAR- techniques with NSBAR- “control technique” have not
been reported before.
In the present version of the manuscript, we now added this information in the text and also provided an new explanatory figure (figure 1) to elucidate the value of this dMUD parameter.
See introduction, page 2, lines 85-89.
Reviewer 1: Questions and Answers
The manuscript needs substantial improvements. To name a few:
1) Much reference is made in the manuscript to supplementary data and to the historical manuscript; however, for better understanding, the main methods and facts must be included in the manuscript itself.
In accordance with the reviewer, we have now included the supplementary text in the manuscript.
For included Tables and text blocks, see methods, page 3, line 102 to page 9 line 292.
2) The 2 tables must appear in the manuscript.
In accordance with the reviewer, we now included the Supplementary Table I as Table I in the manuscript.
3) While the selection of postnatally operated infants is clearly stated (matched selection from the routinely operated Groningen group), the indication for intrauterine surgery in Poland and Bonn and the selection into the study for the open and endoscopic group is completely unclear. A strong selection bias in favor of fetuses with better prognosis must be assumed.
We thank the reviewer for this important point, potentially leading to incorrect assumptions by readers. We now added information on the patient inclusion in the methods. From this information, it can now be understood that a strong selection bias cannot be assumed to be present. This is also indirectly reflected by the similarly high variation in anatomical lesions in all 3 groups, representing the heterogeneity of lesions in SBA.
For information on patient inclusion, See methods, “Patient Inclusion”: page 3 line 121 to page 4 line 121
Additionally, the study protocol also included 3 pre-cautious steps to avoid unwanted potentially confounding effects:
I/ The potential effect by theoretical inclusion of more “favorable” inclusion of lesions in one center compared to the other is compensated by expressing all study –effects against individually age- and lesion matched neonatally operated controls.
II/ By including dMUD as primary study parameter. Before inclusion, the fetal surgical center is unaware of the value in each child.
III/ By the agreement with the fetal and neonatal surgeons in all 3 centers to refrain from any active involvement with data assessment, data processing, and interpretation of the results.
This information is provided in the methods-section, page 3 to page 8
4) How was the anatomical MMC level determined prenatally? Ultrasound? MRI? Was this the same in all children?
This information is now removed from the supplementary file to the methods.
In accordance with the literature, the level of the lesion is taken at the upper border of the MMC, as determined by fetal ultrasonography and confirmed by postnatal MRI.
See methods, page 6, lines 169-173.
5) Which of the data were collected retrospectively from the records and which prospectively according to an established protocol?
All data from the fetal treatment groups were obtained at the outpatient clinic of each center according to the protocol from Groningen. All data from the neonatal treatment group were already obtained and stored according to our own protocol, including neonatal MRI of the brain and spinal cord, neurological investigations (by the same investigators), video-recordings and muscle ultrasound assessments.
This is explained under methods, page 6, 163-168.
6) The intrauterine operated children were all delivered by cesarean section, but the postnatal operated children were delivered vaginally. There are some old retrospective and open data from Canada (but based on hundreds of cases) suggesting that cesarean section before the onset of uterine contractions spares 2 neurologic levels. The authors argue that this aspect is not important for their comparison between open and endoscopic surgery - but it is clearly important for their comparison with postnatally operated cases. And I strongly doubt that their own comparison of only 13 pairs of vaginally delivered vs. caesarean-delivered infants, which they cite in their historical publication, is valid statistical evidence that there is no difference in outcome.
We thank the reviewer for this remark. We agree that more information might be informative. Before conducting the study, we had checked on this since we were aware of the “old” retrospective literature. However, although we thoroughly checked for this point (also in our own database), we did not find scientific substantiation for the indicated extend of a potentially confounding effect by vaginal delivery. This is explained from 3 different perspectives:
- In perspective of literature:
The controversy, as mentioned by the reviewer, refers to old work from Shurtleff and Luthy. These outcomes, however, have received major methodological criticism (including lack of standardization). In subsequent studies, the results could not be confirmed by others. Striving for an evidence-based decision regarding the delivery mode, Greene et al. therefore more recently analyzed 72 SBA children at the age of 2 years. Again, these authors did not observe a beneficial effect from caesarean section on motor function.
- In perspective dMUD analysis:
Preceding to our previously published studies on the effect by fetal intervention, we have also checked and investigated the effect by the delivery modus on muscle ultrasound parameters in our own database. In this study, we compared dMUD results in 13 matched pairs of neonatally operated children born after vaginal delivery versus caesarean section. As indicated in the introduction, dMUD values provide the opportunity to directly quantify the 2nd hit of damage by the MMC. If there would be a difference of 2 spinal segments after caesarean section, this should certainly be quantifiable by the dMUD parameter. However, we observed no difference between both delivery groups.
- In perspective of the neurologic outcome data from the neonatally operated cohort:
Finally, comparing the motor level against the anatomical level of the MMC in our patient group also revealed no differences between the delivery modes. The vaginally delivered patients revealed a median difference of 0 segments (range -2.0 to + 2.0 segments). The patients delivered by caeserean section revealed a median of 0 segments (range -2.0 to 2.0 segments).
In the present version of the manuscript, we have now provided this information in the methods, page 6, lines 135-138 and also in the discussion section, page 15 (Third, we are aware….), line 519-530.
7) Regarding the dMUD method, it is difficult to understand why the gluteus maximus is used as a non-affected reference in cases with low lumbar spine and sacral lesion, since the innervation of the gluteus comes from S1 (and the gluteus medius from L5), so both are affected in many of these cases.
This assumption is incorrect, indeed. As indicated in the methods, we used the Quadriceps muscle as the muscle cranial to the MMC (low lumbar spine lesions) and in higher lesions (thoracic or high lumbar lesions), we applied the Biceps muscle as the muscle cranial to the MMC.
See also methods, page 7, lines 217-220.
8) As the detailed results in Fig. 3 show, the variability of anatomic lesions and functional segments is extremely high in all treatment groups. This is not apparent to the reader in the statistical values or in the boxplots. I therefore recommend that all data be presented in the form of Figure 3 and, in addition, that the Supplementum lists the data for all patients individually.
In a non-biased SBA study group the variation in anatomical lesions is expected to be high. This was confirmed by our results (see also question 3) and provided us the possibility to stratify the dMUD results for the lesional levels.
The variation of lesions is clearly indicated in figure 3 and the levels of the matched pairs are also specifically indicated on the horizontal axis of figure 4 and 5 (both in a and b).
9) Given the small number of cases and the probable selection bias,…...
For the answer, we refer to our response to point 3 (see above)
the statistical analysis can only be descriptive and not confirmatory, as would be possible in a prospective and well-powered study. This should be clearly mentioned.
This study provides descriptive findings on very rare patient groups. After statistical testing, the results can be interpreted as indicative. However, we are aware that a randomized control trial with large numbers of patients is preferrable.
This information is clearly included in the abstract, page 1, lines 37-39 and also well-mentioned in the discussion (3x): - page 14, lines 458-459 and 509-511 and page 15, lines 548-549.
10) Have the statistical tests been corrected for multiple comparisons?
Multiple comparisons arise when a statistical analysis involves multiple simultaneous statistical tests, each of which has a potential to produce a new "discovery”. In the present study, however, we directed our comparisons on the research question whether there is a difference in segmental outcome (reflecting the second hit of damage) between 3 treatment groups. In small descriptive studies, applying directed comparisons to elucidate the research question, testing for multiple comparisons is not used.
In the methods, we now include the descriptive nature of the study in the first sentence of the “statistical analysis”- section” on page 8, line 294.
11) The attempt to calculate a "gain in walking ability" on the basis of motor levels sorted into 3 groups is not convincing. It does not take into account the large variability, so that in individual cases the 20% improvement in walking ability says nothing about the actual walking ability.
For the answer, we refer to our introduction-response above. The secondary aim was to explore the potential neurological significance of the primary outcome (i.e. altered 2nd hit of damage by the fetal intervention in perspective of age- and lesion matches with control values) between both fetal treatment groups. For this purpose, the theoretical prediction based on expectations from rehabilitation guidelines provides more information on the potential theoretical gain, than actual outcomes on ambulation. This is so, because the latter can be subject to the heterogeneity of clinical circumstances.
However, on request of the reviewer, we additionally explored actual functional outcomes on ambulation, as well (see answer to question 13).
And is "walking in long orthoses" really a type of mobility a treatment aim that can be used on the playground or while shopping with the family?
We did not stratify for that since we wanted to deduce the likelihood of functional gain from existing parameters, included in the national rehabilitation guidelines for spina bifida patients (composed by the Dutch society for rehabilitation medicine and Dutch Society for pediatric neurology).
The question whether this is also considered personally relevant for playing and shopping is beyond the aim of our study. However, on request of the reviewer, we additionally explored for actual functional outcomes on ambulation after exclusion of long orthoses, as well (see for further explanation, answer to question 13).
See methods, page 14 methods-section, “tertiary outcomes”, page 8, lines 283-292 and see also results, page 13, lines 429-437.
12) The number of intrauterine and postnatal complications in the endoscopically operated group is extremely high and hardly acceptable. In reference (8), the same group of surgeons, after moving from Bonn to Giessen, reported a much larger number of endoscopically operated cases, with much fewer complications and no intrauterine mortality. This is only briefly mentioned in the manuscript, but is a very important aspect of the Bonn data that should be discussed very clearly.
We agree with the reviewer that this is an important aspect. We also agree that the previously reported complications (as clearly mentioned by us) were extremely high. As the surgical team has now mentioned a strong learning curve in this respect, we had discussed this and made a reference to it.
See discussion, page 13, lines 453-455 and page 14, lines 503-506.
However, since these newly published study data are not associated with the included patients from our study, they are not involved in the present study analysis. We hope that a future independent RCT can help to elucidate this important point to further extent.
This information is clearly included in the abstract, page 1, lines 37-39 and also well-mentioned in the discussion (3x): - page 14, lines 458-459 and 509-511 and page 15, lines 548-549.
13) The authors of reference (8) followed their patients until 30 months of age and can therefore present real data on ambulation. This should also be possible in the present manuscript, as the youngest patients underwent surgery in 2018 and it should be easy to obtain this information (and the patients from the endoscopic group are still in their teens).
As requested by the reviewer, we now added a tertiary aim to compare the actual percentage of functional ambulation between the treatment groups. However, as already mentioned in our response to question 11, actual outcomes can be subject to the heterogeneity of clinical circumstances and theoretically also to the subject reporting the outcome.
See methods, page 14 methods-section, “tertiary outcomes”, page 8, lines 283-292 and see also results, page 13, lines 429-437.
14) It could be a very important finding that the children with an L3 level (anatomically? functionally?) benefited the most from the surgery.
As explained in the methods, this concerns children with an anatomical MMC lesion with a cranial border at level at L3 (i.e. more cranial to L4 and more caudal to L2). We agree that this could be important information when anticipating on the potential effect from fetal surgery in a patient. However, as indicated in the text above, this should be further validated in a randomized controlled trial.
This information is clearly included in the abstract, page 1, lines 37-39 and also well-mentioned in the discussion (3x): - page 14, lines 458-459 and 509-511 and page 15, lines 548-549.
15) It is not clear whether by "segmental gain" the authors mean the difference between anatomic and functional level or the difference between functional level in intrauterinely and postnatally operated infants.
In the methods, we indicate all parameters that refer to segmental gain.
See parameters mentioned under “primary outcome parameters on segmental neurologic function” see methods, page 6, line 178.

Reviewer 2 Report
Sival et al. have aimed to assess neurologic outcomes between fetal-open (OSBAR), fetal-endoscopic (FSBAR) and neonatal (NSBAR) intervention techniques. They showed that OSBAR- and FSBAR-techniques achieved similar neuroprotective results in spina bifida aperta. Moreover, they concluded that a randomized controlled trial could be helpful to reveal and compare ongoing effects by surgical learning curves. The study is orginal, well prepared and discused. In my opinion this paper has a potential to make a positive contribution to the co-existing literature. I have only some minor recommendations for the authors:
1- I think that serum proprotein convertase subtilisin/kexin type 9 (PCSK9) may be involved in the etiopathogenesis of open neural tube defects (NTDs) at the critical steps of fetal neuronal differentiation. Although it has limitations, PCSK9 may be used as an additional biomarker for the screening of NTDs. The study mentioned below may contribute to this article.
“Erol SA, Tanacan A, Firat Oguz E, Anuk AT, Goncu Ayhan S, Neselioglu S, Sahin D. A comparison of the maternal levels of serum proprotein convertase subtilisin/kexin type 9 in pregnant women with the complication of fetal open neural tube defects. Congenit Anom (Kyoto). 2021 Sep;61(5):169-176. doi: 10.1111/cga.12432. Epub 2021 Jun 23. PMID: 34128273.”
Author Response
Reviewer 2
Sival et al. have aimed to assess neurologic outcomes between fetal-open (OSBAR), fetal-endoscopic (FSBAR) and neonatal (NSBAR) intervention techniques. They showed that OSBAR- and FSBAR-techniques achieved similar neuroprotective results in spina bifida aperta. Moreover, they concluded that a randomized controlled trial could be helpful to reveal and compare ongoing effects by surgical learning curves. The study is orginal, well prepared and discused. In my opinion this paper has a potential to make a positive contribution to the co-existing literature. I have only some minor recommendations for the authors:
1- I think that serum proprotein convertase subtilisin/kexin type 9 (PCSK9) may be involved in the etiopathogenesis of open neural tube defects (NTDs) at the critical steps of fetal neuronal differentiation. Although it has limitations, PCSK9 may be used as an additional biomarker for the screening of NTDs. The study mentioned below may contribute to this article.
“Erol SA, Tanacan A, Firat Oguz E, Anuk AT, Goncu Ayhan S, Neselioglu S, Sahin D. A comparison of the maternal levels of serum proprotein convertase subtilisin/kexin type 9 in pregnant women with the complication of fetal open neural tube defects.”
We thank the reviewer for this remark. We have now added this information to the discussion section, page 14, page 456-457.

Round 2
Reviewer 1 Report
The response of the authors to my comments is adequate.
I beg for pardon to have confused quadriceps and gluteus muscles in my question on the intraindividual muscle ultrasound measurements.